# Biological Activity of NHC-Gold-Alkynyl Complexes Derived from 3-Hydroxyflavones

**DOI:** 10.3390/pharmaceutics14102064

**Published:** 2022-09-27

**Authors:** Inés Mármol, Javier Quero, Paula Azcárate, Elena Atrián-Blasco, Carla Ramos, Joana Santos, María Concepción Gimeno, María Jesús Rodríguez-Yoldi, Elena Cerrada

**Affiliations:** 1Departamento de Química Inorgánica, Instituto de Síntesis Química y Catálisis Homogénea-ISQCH, Universidad de Zaragoza-C.S.I.C., Pedro Cerbuna 12, 50009 Zaragoza, Spain; 2Departamento de Farmacología y Fisiología, Medicina Legal y Forense, Unidad de Fisiología, Universidad de Zaragoza, CIBERobn, IIS Aragón, IA2, 50013 Zaragoza, Spain; 3Instituto de Nanociencia y Materiales de Aragón (INMA), CSIC-Universidad de Zaragoza, 50009 Zaragoza, Spain; 4Escola Superior de Tecnologia e Gestão, Instituto Politécnico de Viana do Castelo, Avenida do Atlântico No. 644, 4900-348 Viana do Castelo, Portugal

**Keywords:** Alkynyl, *N*-heterocyclic carbene, gold complexes, hydroxyflavones, antibacterial, anticancer, dihydrofolate reductase, thioredoxin reductase

## Abstract

In this paper we describe the synthesis of new *N*-heterocyclic carbene (NHC) gold(I) derivatives with flavone-derived ligands with a propargyl ether group. The compounds were screened for their antimicrobial and anticancer activities, showing greater activity against bacteria than against colon cancer cells (Caco-2). Complexes [Au(L2b)(IMe)] (**1b**) and [Au(L2b)(IPr)] (**2b**) were found to be active against both Gram-positive and Gram-negative strains. The mechanism of action of **1b** was evaluated by measurement of thioredoxin reductase (TrxR) and dihydrofolate reductase (DHFR) activity, besides scanning electron microscopy (SEM). Inhibition of the enzyme thioredoxin reductase is not observed in either *Escherichia Coli* or Caco-2 cells; however, DHFR activity is compromised after incubation of *E. coli* cells with complex **1b**. Moreover, loss of structural integrity and change in bacterial shape is observed in the images obtained from scanning electron microscopy (SEM) after treatment *E. coli* cells with complex **1b**.

## 1. Introduction

*N*-heterocyclic carbenes (NHC) have been considered as alternatives to phosphines and behave as very good σ-donor ligands, stablishing strong bonds with numerous metals and thus leading to stable NHC complexes for almost all transition metals [1,2,3,4,5,6]. These complexes found application in many fields of research, such as catalysis [7,8,9,10,11], materials and medicine. NHC metal complexes are emerging as novel metallodrugs, due to the high chemical and thermal stability, and many examples have been reported as potential antitumor drugs in the past decade [12,13,14,15,16,17,18,19,20]. Specifically, for gold, many NHC gold complexes represent a promising class of metal based drugs for the treatment of infectious diseases [19,21,22,23,24] or cancer [14,19,25,26,27,28], mainly due to their high stability under physiological conditions and towards blood thiols [29,30]. The use of organometallic derivatives, including NHC complexes, as antibacterial agents has received increasing interest mainly due to their specific modes of action that could hinder the development of mechanisms of bacterial resistance [23,31,32].

The interest in gold derivatives as antimicrobial agents dates back to the end of 19th century, when Koch demonstrated the activity of K[Au(CN)_2_] against *Mycobacterium tuberculosis*. Since then, the research in antibacterial gold derivatives has increased and centred mainly on gold(I) phosphine derivatives and gold(I) *N*-heterocyclic carbenes [23,33]. Among all of them, auranofin (2,3,4,6-tetraacetyl-1-thio-β-D-glucopyranosato-S-triethylphosphane gold(I), Figure 1) stands out. Auranofin, a drug approved in the treatment of rheumatoid arthritis, has shown a potential for new applications in some cancers, parasitic and bacterial infections, HIV and neurodegenerative disorders (Alzheimer’s and Parkinson’s) [33,34,35,36]. Antibacterial properties of auranofin have been widely studied, displaying high efficacy mainly against Gram-positive bacteria, including multidrug resistant strains [24,37,38,39,40]. This effect might be dependent of the affinity of this drug for the redox enzyme thioredoxin reductase (TrxR), which is involved in the thioredoxin system. The thioredoxin system is the only antioxidant defense system of some Gram-positive bacteria such as *Helicobacter pylori* or *Mycobacter tuberculosis*, thus being especially sensitive to auranofin [24,41,42,43]. However, Gram-negative bacteria are less sensitive to gold-containing drugs because of the existence of alternative antioxidant defence systems (e.g., glutathione system, catalase enzyme) [41]. On the other hand, the lack of antibacterial effect against Gram-negative strains is also tentatively attributed to a reduced drug uptake, due to the lack of permeability conferred by the outer membrane barrier [44]. A recent report reveals that a modification in the ligands of auranofin modulate the antimicrobial activity and the resulting complexes are active even against Gram-negative strains [45].

Taking into account this background, here we report on the synthesis of new gold(I) *N*-heterocyclic carbene derivatives with flavone-based ligands functionalized with a propargyl ether group. These complexes are analogous with phosphines, previously described by us, which were shown to be active against Caco-2/TC7 human colon cancer cells through a multitarget mechanism of action [46]. The study of their biological effect reveals that the introduction of NHC ligands induces a reduction in their antitumor activity in favor of antibacterial activity, even against Gram-negative bacteria. Therefore, the aim of the present work was to study the biological activity of NHC-gold-alkynyl complexes derived from 3-hydroxyflavones in relation to the treatment of bacteria-induced diseases and cancer, including the study of the possible mechanism of action against bacteria.

## 2. Materials and Methods

### 2.1. General

All reactions were performed under air atmosphere and solvents were used as received without purification or drying. 3-hydroxyflavones and the corresponding flavone-derived ligands (**La**–**d**) [46], [IPr-H]Cl [47] and [AuCl(NHC)] [48] were prepared according to published procedures. NMR experiments were recorded on a Bruker Avance 400 (400 MHz for ^1^H; 100.62 MHz for ^13^C) or on a Bruker Avance II 300 spectrometer (300 MHz for ^1^H; 75.4 MHz for ^13^C) with chemical shifts (δ, ppm) reported relative to the solvent peaks of the deuterated solvent. MALDI mass spectra were measured on a Micromass Autospec spectrometer in positive ion mode using DIT or DCTB as matrix. Mass spectra were recorded on a BRUKER ESQUIRE 3000 PLUS (Bruker, Boston, MA, USA), with the electrospray (ESI) technique. IR spectra were recorded on a Perkin Elmer Spectrum One FT-IR spectrometer. The samples were measured in solid state using an ATR accessory which covers a wavelength range from 4000 to 200 cm^−1^. UV-Vis spectra were recorded on a Thermo Scientific Evolution 600 spectrophotometer from solutions in quartz cuvettes (1 cm optical path).

### 2.2. Synthesis of the [Au(L)(NHC)] Complexes

To a solution of KOH (0.35 mmol) in MeOH (ca. 10 mL) containing **La**–**d** (0.2 mmol) was added [AuCl(NHC)] (NHC = IMe; IMe = C_5_H_10_N_2_; IPr, IPr = C_27_H_38_N_2_) (0.18 mmol). The corresponding solution was evaporated to dryness after 8 h of stirring at room temperature. Dichloromethane (10 mL) was added and the solution was filtered over Celite, concentrated and the addition of diethyl ether led to white solids, which were isolated by filtration and dried in air. Using this method, the following complexes were prepared:



[Au(La)(IMe)] (R = H; IMe = C_5_H_10_N_2_) (**1a**). Yield: 45%. White solid. ^1^H NMR (300 MHz, CDCl_3_): δ(ppm) = 8.33 (dd, J = 8.4, 1.4 Hz, 2H, H^2′,6′^), 8.29 (dd, J = 8.0, 1.4 Hz, 1H, H^6^), 7.65 (ddd, J = 8.6, 7.1, 1.7 Hz, 1H, H^8^), 7.52–7.43 (m, 4H, H^9^ + H^3′,4′,5′^), 7.37 (ddd, J = 8.1, 7.1, 1.0 Hz, 1H, H^7^), 6.87 (s, 2H, Imidazole), 5.20 (s, 2H, H^2^), 3.79 (s, 6H, Me). ^13^C{^1^H} NMR (75.4 MHz, CDCl_3_): 183.6 (C^2^, Imidazole), 175.2 (C=O), 155.3, 139.4, 133.1, 131.4, 130.2, 129.2 (Ph), 128.2, 126.0, 121.5 (Imidazole), 117.85 (Ph), 98.2 and 98.7 (C≡C), 61.18 (CH_2_), 38.5 (CH_3_). IR: ν(C≡C) 2130 cm^−^^1^, ν(C=O) 1618 cm^−^^1^. MALDI^+^-MS (*m*/*z* (%)): 821 (20, [M + AuIMe]^+^). HMRS (ESI) *m*/*z* calcd. for C_23_H_20_AuN_2_O_3_ 569.1134; found: 569.1130.

[Au(Lb)(IMe)] (R = Br; IMe = C_5_H_10_N_2_) (**1b**). Yield: 52%. White solid. ^1^H NMR (300 MHz, CDCl_3_): δ(ppm) = 8.28 (m, 3H, H^6^+ H^2′,6′^), 7.67–6.68 (m, 3H, H^3′,5′^ + H^8^), 7.53 (d, J = 8.5 Hz, 1H, H^9^), 7.40 (ddd, J = 8.1, 7.1, 1.1 Hz, 1H, H^7^), 6.89 (s, 2H, Imidazole), 5.20 (s, 2H, CH_2_), 3.79 (s, 6H, Me). ^13^C{^1^H} NMR (75.4 MHz, CDCl_3_): δ(ppm) 185.4 (C^2^, Imidazole), 174.7 (C=O), 155.2, 133.3, 131.4, 130.8, 125.9, 124.6 (Ph), 121.6 (Imidazole), 117.8 (Ph), 95.5 (C≡C), 61.0 (-CH_2_-), 37.8 (-CH_3_). IR: ν(C≡C) 2130 cm^−^^1^, ν(C=O) 1613 cm^−^^1^. MALDI^+^-MS (*m*/*z* (%)): 938,9 (100, [M + AuIMe]^+^). HMRS (ESI) *m*/*z* calcd. for C_23_H_19_AuBrN_2_O_3_ 647.0239; found: 649.0186

[Au(Lc)(IMe)] (R = Cl; IMe = C_5_H_10_N_2_) (**1c**). Yield: 47%. White solid ^1^H NMR (300 MHz, CDCl_3_): δ(ppm) = 8.35 (d, J = 1.9 Hz, 2H, H^2′,6′^), 8.29 (dd, J = 8.0, 1.5 Hz, 1H, H^6^) 7.68 (ddd, J = 14.3, 8.0, 4.5 Hz, 1H, H^8^), 7.56–7.47 (m, 3H, H^3′,5′^ + H^9^), 7.41 (m, 1H, H^7^), 6.89 (s, 2H, Imidazole), 5.20 (s, 2H, CH_2_), 3.80 (s, 6H, Me). ^13^C{^1^H} NMR (75.4 MHz, CDCl_3_): δ(ppm) 186.8 (C^2^, Imidazole), 175.1 (C=O), 155.2, 136.3, 133.2, 130.6, 128.5, 125.8, 124.6 (Ph), 121.7 (Imidazole), 118.0 (Ph), 95.5 (C≡C), 61.1 (CH_2_), 37.8 (CH_3_). IR: ν(C≡C) 2130 cm^−^^1^, ν(C=O) 1631 cm^−^^1^. MALDI^+^-MS(*m*/*z* (%)): 895.1 (100, [M + AuIMe]^+^). HMRS (ESI) *m*/*z* calcd. for C_23_H_19_AuClN_2_O_3_ 603.0744; found: 603.0746.

[Au(Ld)(IMe)] (R = OMe; IMe = C_5_H_10_N_2_) (**1d**). Yield: 44%. White solid. ^1^H NMR (300 MHz, CDCl_3_): δ(ppm) = 8.34 (d, J = 9.1 Hz, 2H, H^2′,6′^), 8.27 (dd, J = 8.0, 1.4 Hz, 1H, H^6^), 7.65 (ddd, J = 8.6, 7.0, 1.7 Hz, 1H, H^8^), 7.51 (dd, J = 8.4, 0.6 Hz, 1H, H^9^), 7.38 (ddd, J = 8.1, 7.1, 1.1 Hz, 1H, H^7^), 7.04 (d, J = 9.1 Hz, 2H, H^3′,5′^), 6.88 (s, 2H, Imidazole), 5.16 (s, 2H, CH_2_), 3.89 (s, 3H, OCH_3_) 3,78 (s, 6H, Me). ^13^C{^1^H} NMR (75.4 MHz, CDCl_3_): δ(ppm) = 183.1 (C^2^, Imidazole), 172.4 (C=O), 161.6, 155.2, 133.3, 130.7, 125.8, 124.7, 124.0 (Ph), 121.6 (Imidazole), 117.9, 113.8 (Ph), 97.5 (C≡C), 59.1 (CH_2_), 55.4 (OMe), 37.5 (CH_3_). IR: ν(C≡C) 2130 cm^−^^1^, ν(C=O) 1605 cm^−^^1^. MALDI^+^-MS (*m*/*z* (%)): 891 (100, [M-AuIMe]^+^). HMRS (ESI) *m*/*z* calcd. for C_24_H_22_AuN_2_O_4_ 599.1245; found: 599.1263.



[Au(La)(IPr)] (R = H; IPr = C_27_H_38_N_2_) (**2a**). Yield: 45%. White solid. ^1^H NMR (300 MHz, CDCl_3_): δ(ppm) 8.22 (dd, J = 8.0, 1.6 Hz, 1H, H^6^), 8.15–8.12 (m, 2H, H^2′,6′^), 7.64 (ddd, J = 15.6, 7.2, 1.7Hz, 1H, H^8^), 7.52–7.23 (m, 7H, H^3′,5′^ + H^4^ + H + H^9^ + IPr), 7.09 (s, 2H, Imidazole), 5.20 (s, 2H, CH_2_), 2.5 (sept, 4H, CHMe_2_), 1.25 and 1.20 (d, J = 6.9 Hz, 24 H, CHMe_2_). ^13^C{^1^H} NMR (75.4 MHz, CDCl_3_): 197.6 (C^2^, Imidazole), 174.8 (C=O), 145.5, 130.4, 129.1, 127.9, 127.05, 124.0, 123.9 (Ph), 123.0 (Imidazole), 117.7 (Ph), 61.1(CH_2_), 28.7 (CHMe_2_), 24.51 (CH_3_), 23.95 (CH_3_). IR: ν C≡C) 2130 cm^−^^1^, ν(C=O) 1620 cm^−^^1^. MS (MALDI+, *m*/*z* (%)): 861.3 (35, [M]^+^), 835 (40, [M-2CH_3_] ^+^), 1445 (50, [M + AuIPr]^+^). HMRS (ESI) *m*/*z* calcd. for C_45_H_48_AuN_2_O_3_ 861.3325; found: 861.3338.

[Au(Lb)(IPr)] (R = Br; IPr = C_27_H_38_N_2_) (**2b**). Yield: 45%. White solid. ^1^H NMR (300 MHz, CDCl_3_): δ(ppm) = 8.20 (dd, J = 8, 1.4 Hz, 1H, H^6^), 8.08 (d, J = 8.8 Hz, 2H, H^2′,6′^), 7.69–7.62 (m, 1H, H^8^), 7.49–7.37 (m, 7H, H^3′,5′^ + H^4^ + H^8^ + H + IPr), 7.11 (s, 2H, Imidazole), 5.06 (s, 2H, CH_2_), 2.52 (sept, 4H, CHMe_2_), 1.25 and 1.20 (d, J = 6.9 Hz, 24H, CHMe_2_). ^13^C{^1^H} NMR (75.4 MHz, CDCl_3_): 190.9 (C^2^, Imidazole), 175.0 (C=O), 155.1, 145.5, 134.1, 133.0, 131.2, 130.5, 130.1, 127.6, 126.0, 124.3, 124.1 (Ph), 123.0 (Imidazole), 117.7 (Ph), 97.6 (C≡C), 61.4 (CH_2_), 28.7 (CHMe_2_), 24.5 (CH_3_), 23.9 (CH_3_). IR: ν(C≡C) 2130 cm^−^^1^, ν(C=O) 1606 cm^−^^1^. MS (MALDI^+^, *m*/*z* (%)): 939 (18, [M]^+^), 1524.9 (100, [M + AuIPr]^+^). Elemental analysis calcd. (%) for C_45_H_47_AuBrN_2_O_3_ (Pm = 940.2436) C, 57.45; H, 5.04; N, 2.98, found: C: 56.73; H: 5.00; N 2.95. HMRS (ESI) *m*/*z* calcd. for C_45_H_47_AuBrN_2_O_3_ 940.2430; found: 941.2388.

[Au(Lc)(IPr)] (R = Cl; IPr = C_27_H_38_N_2_) (**2c**). Yield: 52%. White solid. ^1^H NMR (300 MHz, CDCl_3_): δ(ppm) = 8.20 (dd, J = 8, 1.4 Hz, 1H, H^6^), 8.06 (d, J = 8.8 Hz, 2H, H^2′,6′^), 7.66 (m, 1H, H^8^), 7.42–7.02 (m, 10H, H^7^ + H^9^ + H^2′,6′^ + IPr), 7.11 (s, 2H, Imidazole), 5.05 (s, 2H, CH_2_), 2.49 (sept, 4H, CHMe_2_), 1.25 and 1.20 (d, J = 6.9 Hz, 24H, CHMe_2_). ^13^C {^1^H} NMR (75.4 MHz, CDCl_3_): δ(ppm) = 190.1 (C^2^, Imidazole), 178.4(C=O), 145.5, 134.0, 132.3, 130.5, 128.3, 126.0,124.3, 124.1 (Ph) 123.0 (Imidazole), 61.4 (CH_2_), 28.7 (CHMe_2_), 24.5 (CH_3_), 23.9 (-CH_3_). IR: ν C≡C) 2130 cm^−^^1^, ν(C=O) 1633 cm^−^^1^. MS (MALDI^+^, *m*/*z* (%)): 835 (70, [M-4CH_3_]^+^), 895 (31, [M]^+^), 1479 (100, [M + AuIPr]^+^). Elemental analysis calcd. (%) for C_45_H_47_AuClN_2_O_3_ (Pm = 895.2935) C, 60.30; H, 5.29; N, 3.13, found: C: 60.13; H: 5.14; N 2.68. HMRS (ESI) *m*/*z* calcd. for C_45_H_47_AuClN_2_O_3_ 895.2935; found: 895.2911.

[Au(Ld)(IPr)] (R = OMe; IPr = C_27_H_38_N_2_) (**2d**). Yield: 74%. White solid. ^1^H NMR (300 MHz, CDCl_3_): δ(ppm) = 8.21 (dd, J = 8.0, 1.4 Hz, 1H, H^6^) 8.14 (m, J = 8.0, 2H, H^2′,6′^), 7.64 (ddd, J = 7.1, 4.9, 1.7 Hz, 1H, H^8^), 7.5–7.22 (m, 8H, H^7^ + H^9^ + IPr), 7.10 (s, 2H, Imidazole), 6.8 (m, 2H, H^3′,5′^), 5.00 (s, 2H, CH_2_), 2.5 (m, 4H, CHMe_2_), 1.25 and 1.20 (d, J = 6.9 Hz, 24H, CHMe_2_). ^13^C{^1^H} NMR (75.4 MHz, CDCl_3_): δ (ppm) =190.6 (C^2^, Imidazole), 175.0 (C=O), 161.1, 155.0, 145.5, 134.1, 132.6, 130.7, 130.4, 125.9, 124.3, 124.0 (Ph), 123.0 (Imidazole), 117.6, 113.5 (Ph), 97.6 (C≡C), 61.1(CH_2_), 55.3 (OCH_3_), 28.7 (CHMe_2_), 24.5(-CH_3_), 24.0 (CH_3_). IR: ν(C≡C) 2130 cm^−^^1^, ν(C=O) 1605 cm^−^^1^. MS (MALDI^+^, *m*/*z* (%)): 891 (40, [M]^+^), 1475(100, [M + AuIPr]^+^). Elemental analysis calcd. (%) for C_46_H_50_AuN_2_O_4_ (Pm = 891.3431) C, 61.95; H, 5.65; N, 3.14, found: C: 61.77; H: 5.87; N 3.12. HMRS (ESI) *m*/*z* calcd. for C_46_H_50_AuN_2_O_4_ 891.3431; found: 891.3437.

### 2.3. Distribution Coefficient (LogP)

The n-octanol-water coefficients of the complexes were determined as previously reported [49] using a shake-flask method. Buffered-saline distilled water (100 mL, phosphate buffer [PO_4_^3−^] = 10 mM, [NaCl] = 0.15 M, pH 7.4) and n-octanol (100 mL) were shaken for 72 h to allow saturation of both phases. Approximately 1 mg of the complexes was dissolved in 5 mL of the aqueous phase and 5 mL of the organic phase were added, mixing for 10 min. The resulting emulsion was centrifuged to separate the phases. The concentration of the compounds in each phase was determined using UV absorbance spectroscopy. LogP was defined as log{[compound(organic)]/[compound(aqueous)]}.

### 2.4. Stability

The stability of the gold complexes has been analyzed by absorption UV spectroscopy. UV-Vis absorption spectra of the complexes were recorded on a Thermo Scientific spectrophotometer. Briefly: solutions of the new derivatives (5 × 10^−^^5^ M) in PBS (pH = 7.4) were prepared from 10 mM DMSO stock solutions of the complexes and thereafter monitored measuring their electronic spectra over 24 h after incubation at 37 °C.

### 2.5. Disk Diffusion Test 

To evaluate the antibacterial potential of gold derivatives, six bacterial strains were selected: *E. coli, P. aeruginosa*, *S. enterica*, *E. faecalis*, *L. monocytogenes*, *S. epidermidis*, *S. aureus*. Dis diffusion assay was performed following Clinical & Laboratory Standards Institute (CLSI) [50] recommendations. Active cultures (0.5 McFarland) from all strains were spread onto Mueller–Hinton Agar (MHA, Oxoid, Basingstoke, UK) and disks impregnated with 10 µL of each gold complex or control (DMSO for negative control, phenol for positive control) were placed. Plates were and incubated for 22 h ± 2 h at 37 °C ± 1 °C and then the diameter of the inhibition zone was determined.

### 2.6. Bacterial Growth and MIC Calculation E. coli

ATCC 25922 strain was grown in Luria–Bertani (LB) broth medium (Sigma, L3022-250G, St. Louis, MI, USA) at 37 °C. Minimum inhibitory concentration (MIC) was determined in 96-well plates, adding 100 µL of bacterial suspension at 10^6^ CFU/mL to each well. Then, 100 µL of gold compound at desired concentrations (100, 75, 50, 30, 25, 20, 15 and 0 µM in LB) was added to each well and bacteria were incubated at 37 °C for 24 h. To evaluate bacterial growth, 30 µL of resazurin (0.1 mg/mL in ddH_2_O) was added to each well and the change from blue to pink upon bacterial growth was evaluated after 30 min of incubation at 37 °C.

### 2.7. Dihydrofolate Reductase and TrxR Activities Activity Assay in E. coli

For sample preparation, *E. coli* were lysated by sonication. After ultrasonic homogenization, the lysate was centrifuged 1 min at 13,000 rpm and the pellet was resuspended in extraction buffer. DHFR activity assay was performed adding 1 µL of potential inhibitor at the desired concentration (20 µM trimethoprim was used as positive control) and 50 µM dihydrofolic acid (Sigma-Aldrich, D7006-10MG, St Louis, MA, USA) to 0.01–0.03 mg of cell lysate and completed with reaction buffer to final volume of 100 µL. NADP+ formation was monitored for 1 h measuring absorbance at 420 nm every 30 s with SPECTROstar Nano (BMG Labtech, Ortenberg, Germany). Both extraction and reaction buffer consisted of 10 mM Tris-HCl, 1 mM EDTA, 100 mM NaCl and protease inhibitor.

TrxR activity assay was performed adding 1 µL of potential TrxR inhibitor at the desired concentration, 5mM DTNB 100 µM NADPH to 0.01–0.03 mg of cell lysate and completed with reaction buffer to a final volume of 100 µL. TNB conversion was monitored measuring absorbance at 405 nm every 60 s for 20 min with SPECTROstar Nano (BMG Labtech). Both extraction and reaction buffer coincided with buffer used in DHFR activity assay.

### 2.8. Preparation of E. coli for Scanning Electron Microscopy (SEM)

500 μL of a *E. coli* culture (10^7^ CFU mL^−^^1^) were added per well to a 12-well plate and were further treated with the complex **1b** at its MIC and 1/2× MIC. A positive control of bacteria without any treatment and a negative control of only culture medium were included, following the same protocol as the rest of the conditions. Then, the plate was incubated at 37 °C with mild agitation for 24 h. After this time, the bacterial suspensions were transfered into 1.5 mL centrifugation tubes. Then, the bacteria were washed and fixed using glutaraldehyde 2.5% in phosphate buffer 10 mM at pH 7.2, with 2 h incubation. Then, bacteria were gradually dehydrated using EtOH solutions at different *v*/*v*% (25, 50, 75 and 100). 10 µL of the dehydrated bacteria solution was placed on a 5 × 5 mm Si waffle, left to dry and coated with a thin layer of carbon.

### 2.9. Caco-2 Cell Culture and Treatment

Human colorectal adenocarcinoma Caco-2 cells were kindly provided by Dr. Edith Brot-Laroche (Universite Pierre et Marie Curie-Paris 6 UMR S872, Les Cordeliers, France) and were maintained in a humidified atmosphere of 5% CO_2_ at 37 °C in Dulbecco’s Modified Eagles medium (DMEM) (Gibco Invitrogen, Paisley, UK) supplemented with 20% fetal bovine serum, 1% non-essential amino acids, 1% penicillin (1000 U/mL), 1% streptomycin (1000 mg/mL) and 1% amphotericin (250 U/mL). For treatment, gold complexes were initially dissolved in DMSO to a concentration of 20 mM and then diluted on cell culture without fetal bovine serum to the required work concentrations, which was added 24 h post-seeding. Then, cells were incubated at 37 °C for the required time for each assay.

### 2.10. Antiproliferative Activity and IC_50_ Calculation

Cell viability was determined with the 3-(4,5-dimethyl-2-thiazoyl)-2,5-diphenyltetrazolium bromide (MTT) assay as previously described by Mármol et al. [51]. For IC_50_ calculation, cells were grown in 96-well plates at a density of 4000 cells per well and incubated overnight at standard culture conditions. Then, cells were exposed 72 h to 0–20 μM of each gold complex and changes in absorbance were converted into percentage of growth inhibition in comparison to negative control (DMSO. Absorbance was measured with SPECTROstar Nano (BMG Labtech).

### 2.11. Measurement of Intracellular ROS Levels

Caco-2 cells were grown in 96-well plates (4000 cells/well) and incubated overnight at 5% CO_2_ and 37 °C. Then, cells were exposed to the required concentration of gold complex for 1 h and total intracellular ROS levels determination was performed as previously described by Sanchez-de-Diego et al. [52].

### 2.12. Measurement of TrxR Activity

A measurement of human recombinant thioredoxin reductase was performed as previously described by Quero et al. [53]. Briefly, gold complex was dissolved in DMSO at the desired concentrations and was incubated for 45 min with recombinant human TrxR (SIGMA SRP6081) at 37 °C in a 96 well plate. 25 μL of reaction mix, composed by 500 μL PBS pH 7.4, 80 μL 100 mM EDTA pH 7.5, 20 μL 0.05% BSA, 20 mM 100 μL NADPH, and 300 μL distilled H_2_O, was then added to each well. 25 μL of 20 mM 5,5′-dithiobis(2-nitrobenzoic acid) (DNTB), dissolved in 100% EtOH, were added to start the reaction and changes in absorbance at 405 nm were monitored every 10 s for 6 min. For comparison purposes, the activity of recombinant TrxR in presence of DMSO was used as negative control.

## 3. Results and Discussion

### 3.1. Synthesis

The synthesis of the 3-hydroxyflavones and the corresponding flavone-derived ligands (**La**–**d**) functionalized with the propargyl residue were conducted as previously reported by us [46]. The deprotonation of the alkyne ligands **La**–**d** with potassium methoxide followed by the addition of [AuCl(NHC)] (NHC = 1,3-bis(2,6-di-i-propylphenyl)imidazolidin-2-ylidene (IPr; IPr = C_27_H_38_N_2_) and 1,3-dimethylimidazolidin-2-ylidene (IMe; IMe = C_5_H_10_N_2_) afforded the corresponding gold(I) derivatives **1a**–**d** and **2a**–**d** (Figure 1) as air-stable solids. The formation of the alkynyl carbene species is confirmed by the disappearance of the acetylenic triplet and the downfield shift of the methylene signal in their ^1^H NMR spectra. The signals corresponding to the protons at the C^4^ and C^5^ carbon atoms of the imidazole in the gold complexes are upfield with respect to the starting material [AuCl(NHC)] [54]. Higher differences are observed in their ^13^C{^1^H} NMR being the signals corresponding to the C_carb_ downfield displaced in the new complexes in comparison to the chloride gold derivatives (Δ*δ* of ca. 15 ppm), that could be rationalized as a consequence of the higher donor capacity of the chloride ligand. Besides, the *ν*(C≡C) vibration observed in the free alkyne as a weak band at 2110 cm^−^^1^ in the infrared spectra is displaced to 2130 cm^−^^1^ in the gold complexes, in addition to the disappearance of the band around 3200 cm^−^^1^ corresponding to the terminal alkyne (≡C-H) stretch.

### 3.2. Solution Stability and Lipophilicity

Stability of a complex in solution is essential; consequently, it should not undergo any reaction with oxygen, water, or the excipients present in the solution, which could compromise its activity. We have studied the solution stability of the N-heterocyclic carbene complexes under physiological conditions, after dissolving them in phosphate-buffered saline (PBS) at pH 7.4 and maintained at 37 °C for 24 h. The resulting UV−vis absorption spectra feature bands in the region of 210–350 nm (Appendix A) similar to that observed in the related phosphane complexes previously described by us [46], which remain unchanged in shape or displacement in the absorbance maximum. No red- or blue shifts are observed in the absorbance in addition to the absence of the characteristic absorbance of gold reduction at around 500 nm, implying a certain stability under physiological conditions.

The lipophilicity of a compound is an important physicochemical property of a potential drug. It is generally related to its diffusion and permeation into the lipid bilayer membrane of the cells, thus affecting its activity and toxicity. The lipophilicity of the new derivatives was determined by the partition coefficient water/octanol, log P (Table 2) by measuring the concentration ratio of the complex in octanol and buffered aqueous solution (pH = 7.4) at an equilibrium state. All the complexes display lipophilic character with a wide range of values when comparing the complexes by tetrads of the same NHC ligand (IMe, values in the range 0.34–1.96 vs. IPr, values in the range 0.42–3.4).

### 3.3. Biological Activity

#### 3.3.1. Antibacterial Effect

Given that NHC-gold derivatives might display antibacterial activity, we evaluated the antibacterial effect of the complexes toward a panel of Gram-positive and Gram-negative strains using the disk diffusion method. The selected strains are human pathogen or commensal bacteria and the results are summarized on Table 1.

Complexes **1a**, **1c** and **2d** showed no antibacterial effect; however, complexes **1d** and **2c**, as can be observed in Table 1, displayed the ‘classical’ antibacterial effect of gold derivatives since they were able to inhibit the growth of all Gram-positive bacterial strains, but were inactive against Gram-negative strains. These results suggest that both complexes might inhibit bacterial thioredoxin reductase (TrxR), the most common target for gold compounds, as previously reported for auranofin [44,55,56], as well as other gold complexes and ruthenium derivatives with N-heterocyclic carbene ligands [57,58,59]. Contrarily, complexes **1b** and **2b** were found able to interfere with the development of both Gram positive and Gram negative strains, the effect of the complex coordinated with the carbene IMe ([Au(L2b)(IMe)] (**1b**)) being greater than that detected for its counterpart with IPr ([Au(L2b)(IPr)] (**2b**)).

The observed effect of complexes **1b** and **2b** toward Gram negative strains suggest that these derivatives might be able to interact with a molecular target other than thioredoxin reductase, which is essential for the antioxidant system of Gram-positive bacteria. In fact, incubation of *E*. *coli* cell lysates with gold complex **1b** did not result in a significative reduction of TrxR activity (Figure 2A), so this enzyme was discarded as a potential bacterial target for **1b** antibacterial activity.

One of the major problems with bacteria is drug resistance, the search for new therapeutic targets and inhibitors to overcome the problems of antibiotic resistance being essential. Dihydrofolate reductase (DHFR) is an important enzyme required to maintain bacterial growth and, consequently, DHFR inhibitors can be considered as effective agents for treating bacterial infections [60]. Dihydrofolate reductase plays a key role in folate cycle. This ubiquitous enzyme, found in all organisms, is involved in the biosynthesis of 5,6,7,8-tetrahydrofolic acid from 5,6-dihydrofolic acid by enantiospecific hydride transfer from NADPH cofactor, being essential for the biosynthesis of DNA bases and amino acids [43].

Its inhibition interrupts the supply of 5-Thymidylic acid (conjugate base thymidylate), also known as thymidine monophosphate, thus stopping DNA synthesis. For this reason, targeting DHFR to inhibit cell proliferation has been a common approach in the treatment of different diseases such as cancer, malaria and bacterial infections [60,61,62,63,64]. Since previous studies with related imidazolate gold complexes showed DHFR-inhibition capacity in bacteria [65] or in breast cancer cells [66], *E. coli* lysates were incubated with gold complex **1b** and reduction of dihydrofolate to tetrahydrofolate was monitored, observing that, like trimethoprim (TMP), a well-known DFHR inhibitor and widely prescribed antimicrobial agent **1b** significantly decreased DHFR activity (Figure 2B). These results suggested that DHFR could play an important role in **1b** antibacterial activity.

Furthermore, the 3-hydroxyflavone moiety might be at least partially responsible for the reported antibacterial activity on these strains. Some authors have found that the lipophilic nature of flavones confers on them the ability to disrupt bacterial membrane permeability and, as a consequence, inhibit their growth [67,68]. With this idea, scanning electron microscopy was used to further observe the effect of complex **1b** on cell death and the possible interaction with the membrane of *E. coli* (Figure 3A). Indeed, two characteristic changes in cell morphology can be observed upon treatment with complex **1b**. The most notorious one is a rougher surface of bacteria, with the presence of wrinkles and creases. This change in morphology of *E. coli*, indicative of membrane damage, has previously been observed for naturally occurring flavonoids [69], disinfectants such as chlorhexidine [70] and antimicrobial peptides [71]. Another effect also observed is the shrinking of bacteria, which is concentration dependent (Figure 3B). Complex **1b** could also have an impact on the reduction of biofilm matrix (Figure 3A, red hollow arrows), the extracellular material produced by bacteria in which the biofilm cells are implanted. The biofilm matrix confers important protection to the biofilm bacteria and its production is connected to mechanisms of antimicrobial resistance [72,73]. Thus, it is of interest to develop new antimicrobial complexes with the capacity to decrease the production of biofilm matrix.

#### 3.3.2. Antiproliferative Activity

Since in a previous work, a series of gold derivatives coordinated with 3-hydroxyflavones showed promising anticancer effect [46] we decided to analyze the antiproliferative effect of complexes **1a**–**d** and **2a**–**d** towards undifferentiated cells of the Caco-2 cell line as a model of colorectal adenocarcinoma. IC_50_ values of auranofin were also determined as a positive control for comparison purposes. As summarized in Table 2, all complexes showed much higher IC_50_ values than auranofin, which suggests that the reference drug displays greater anticancer activity than the novel gold(I) derivatives. Therefore, these series of gold derivatives do not highlight due to their anticancer potential.

The use of 1,3-dimethylimidazol-2-ylidene (IMe) results in an increase on the antiproliferative effect of the resultant complex, since the starting material [AuCl(IMe)] is ineffective against Caco-2 cells; in contrast, 1,3-bis(2,6-di-i-propylphenyl)imidazolidin-2-ylidene (IPr) displays the opposite effect. The IMe series (**1a**–**d**) display higher antiproliferative effect than their IPr counterparts **2a**–**d**, except for complexes **1a** and **2a**. In this particular case, the IC_50_ of the complex with the IPr carbene (**2a**) is 2.65 times lower than the homologue with IMe (**1a**), which is indicative of a significantly (*p* < 0.05) higher antiproliferative effect. It is also noticeable that complex **2a** is the most active complex from the IPr series in terms of IC_50_ value.

Table 2 also shows distribution coefficient values of the novel gold(I) derivatives. Whereas complexes **2b**–**d** display a strong lipophilic nature, complex **2a** shows a balanced hydrophilic-lipophilic character. However, its antiproliferative effect is lower than that obtained for other gold (I) complexes derived from 3-hydroxyflavones, previously reported by some of us [46].

Nonlinear correlation was found between the lipophilic character of the compounds, by changing the substituent of the alkyne moiety, and their cytotoxic activity. When increasing lipophilic character, with increasing logP values, the activities of the complexes describe an inverted U-shaped curve (Figure 4). Thus, the highest IC_50_ value should be expected for derivatives with those ligands able to give intermediate logP values, which in the case of IPr carbene derivatives is around 2 and 1 for the IMe counterpart and the highest activity should be found for low lipophilic character.

On the other hand, the lipophilic nature of the complexes might not be the unique origin of their reduced antiproliferative effect obtained. We have previously reported [46] that complex [Au(L2c)PPh_3_], whose distribution coefficient value was 1.17 and similar than those obtained for complex **2c**, displayed an IC_50_ value of 5.34 ± 0.05. Therefore, cellular uptake led by the lipophilic character of the complexes might not explain their poor antiproliferative effect. We hypothesized that the presence of the strong donor NHC and alkynyl ligands might disfavored the dissociation of one of the ligands and thus impeded the interaction of the gold complex and molecular target, thus leading to a poor anticancer effect.

Similar to Gram-positive bacteria, human TrxR is considered the canonical molecular target of gold derivatives with chemotherapeutic potential. Given that we have previously found evidence suggesting that this series of complexes might not display affinity for this redox enzyme (Figure 1), we decided to evaluate the effect of **1b** on the activity of human recombinant TrxR. We found that incubation of recombinant TrxR with 40 μM of **1b** led to no significant changes (*p* > 0.05) in enzymatic activity (Figure 5A). Furthermore, treatment of Caco-2 cells with increasing amounts of [Au(L2b)(IMe)] (**1b**) (10, 20 and 40 μM) resulted in no significant changes (*p* > 0.05) in reactive oxygen species (ROS) levels after 1 h incubation (Figure 5B). Taken together, these results suggest that complex **1b** does not inhibit the activity of TrxR whether human or bacterial. The lack of affinity toward human TrxR might explain the low anticancer effect of this complex in comparison with the counterpart [Au(L2b)PPh_3_], which shares the same ligand derived from hydroxyflavone and was able to inhibit thioredoxin reductase enzyme and induce an increase in ROS levels that triggered apoptotic cell death. Our present results suggest that the replacement of the PPh_3_ ligand by the carbene moiety might lead to an inability to interact with TrxR in both bacteria and cancer cells in favor of DFHR interaction in bacteria.

## 4. Conclusions

In conclusion, here we describe the synthesis of new *N*-heterocyclic carbene (NHC) gold(I) derivatives with flavone-derived ligands with a propargyl ether group similar to the homologous complexes with phosphanes, previously described by us. Although both families of complexes share similar structures, the substitution of the phosphane ligand by the carbene moiety induces a reduction in their antitumor activity in favor of antibacterial activity. Complexes [Au(L2b)(IMe)] (**1b**) and [Au(L2b)(IPr)] (**2b**) were found to be active against both Gram-positive and Gram-negative strains, which points to a different mechanism than that based on enzyme thioredoxin reductase interaction. In fact, no significative reduction of TrxR activity of *E. coli* cell lysates were detected after complex **1b** incubation. However, **1b** significantly decreased dihydrofolate reductase (DHFR) activity, suggesting that DHFR could play an important role in the antibacterial activity induced by [Au(L2b)(IMe)] (**1b**). Scanning electron microscopy (SEM) clearly showed a morphological change in the bacteria *E. coli* envelope after treatment with complex **1b**, which could lead to destabilization of the bacterial membrane causing loss of structural integrity.

## Data Availability

Not applicable.

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
