# Peer review of "Biological Activity of NHC-Gold-Alkynyl Complexes Derived from 3-Hydroxyflavones"

_pharmaceutics, 2022, doi:10.3390/pharmaceutics14102064_

Round 1

Reviewer 1 Report

The manuscript titled “Biological Activity of NHC-Gold-Alkyne Complexes De-rived from 3-hydroxyflavones” written by Inés Mármol, Javier Quero, Paula Azcárate, Elena Atrian, Carla Ramos, Joana Santos, M. Concepción Gi-meno, M. Jesús Rodríguez-Yoldi, and Elena Cerrada concerns the synthesis of new N-heterocyclic carbene (NHC) gold(I) derivatives with flavone-derived ligands with a propargyl ether group. The new compounds were screened for their antimicrobial and anticancer activities, showing greater activity against bacteria than against colon cancer cells (Caco-2).

The article has some confusion which should be eliminated.

The letter N in all sentences N-heterocyclic carbene should be written as italic N-heterocyclic carbene.

From page 3 the spectroscopic data needs attention because NMR is written like this not as the RMN.

Acronym NMR comes from Nuclear magnetic resonance spectroscopy.

For all compounds the elemental analysis should be added or high resolution measurement.

Furthermore, all new compounds should be given the exact elemental composition for easier identification.

The imidazole derivatives should also be included in the chemical formula for a better visualization of chemical structure.

For 1a the CDCl3 should be written as CDCl3.

In the synthesis part I found the letter y in the sentence: The deprotonation of the alkyne ligands La-d with potassium methoxide followed by the addition of [AuCl(NHC)] (NHC = 1,3-bis(2,6-di-i-propylphenyl)imidazolidin-2-ylidene (IPr) y 1,3-dimethylimidazolidin-2-ylidene (IMe)) afforded the corresponding gold(I) derivatives 1a-d and 2a-d (Scheme 1) as air-stable solids. I think it is a mistake, and the word ‘and’ should be there instead of y.

Author Response

The manuscript titled “Biological Activity of NHC-Gold-Alkyne Complexes De-rived from 3-hydroxyflavones” written by Inés Mármol, Javier Quero, Paula Azcárate, Elena Atrian, Carla Ramos, Joana Santos, M. Concepción Gimeno, M. Jesús Rodríguez-Yoldi, and Elena Cerrada concerns the synthesis of new N-heterocyclic carbene (NHC) gold(I) derivatives with flavone-derived ligands with a propargyl ether group. The new compounds were screened for their antimicrobial and anticancer activities, showing greater activity against bacteria than against colon cancer cells (Caco-2).

The article has some confusion which should be eliminated.

  • The letter N in all sentences N-heterocyclic carbene should be written as italic N-heterocyclic carbine

It has been corrected along the text.

  • From page 3 the spectroscopic data needs attention because NMR is written like this not as the RMN. Acronym NMR comes from Nuclear magnetic resonance spectroscopy.

RMN has been changed by NMR in the experimental section

  • For all compounds the elemental analysis should be added or high resolution measurement. Furthermore, all new compounds should be given the exact elemental composition for easier identification.

We have measured high-resolution mass-spectrometry (HRMS-ESI) and written the exact elemental composition in the experimental section.

  • The imidazole derivatives should also be included in the chemical formula for a better visualization of chemical structure.

We have included the chemical formula in the imidazole derivatives.

  • For 1a the CDCl3 should be written as CDCl3.

It has been corrected.

  • In the synthesis part I found the letter y in the sentence: The deprotonation of the alkyne ligands La-d with potassium methoxide followed by the addition of [AuCl(NHC)] (NHC = 1,3-bis(2,6-di-i-propylphenyl)imidazolidin-2-ylidene (IPr) y1,3-dimethylimidazolidin-2-ylidene (IMe)) afforded the corresponding gold(I) derivatives 1a-d and 2a-d (Scheme 1) as air-stable solids. I think it is a mistake, and the word ‘and’ should be there instead of y.

Thanks a lot, we have corrected the mistake.

Reviewer 2 Report

This paper describes synthesis of N-heterocyclic carbene-Au(I)-alkyne complexes containing 3-hydroxyflavone structural motif. Using a simple synthetic procedure, the author obtained the series of such complexes.

The main part of the paper devoted to the study of biologically active properties of these complexes.

As a result, it has been found that the complexes exhibit activity against both Gram-positive and Gram-negative strains.

 This is a very useful paper for medicinal chemists!

 However, there are some comments and recommendations to improve the paper as follows.

- Give structure of auranofin in a special figure in the introduction section of the paper.

- All newly obtained compounds must be characterized by high-resolution mass-spectrometry (HRMS) data. Please, provide HRMS data for all complexes 1 and 2.

- Complexes 1 and 2 are solid compounds. Please, give their melting points.

 After these corrections the paper may be accepted.

Author Response

This paper describes synthesis of N-heterocyclic carbene-Au(I)-alkyne complexes containing 3-hydroxyflavone structural motif. Using a simple synthetic procedure, the author obtained the series of such complexes. The main part of the paper devoted to the study of biologically active properties of these complexes. As a result, it has been found that the complexes exhibit activity against both Gram-positive and Gram-negative strains. This is a very useful paper for medicinal chemists!

However, there are some comments and recommendations to improve the paper as follows.

  • Give structure of auranofin in a special figure in the introduction section of the paper.

The structure of auranofin has been included in a chart at the end of the introduction.

  • All newly obtained compounds must be characterized by high-resolution mass-spectrometry (HRMS) data. Please, provide HRMS data for all complexes 1 and 2.

We have measured the HMRS (ESI) for all the new complexes, which are recorded in the experimental section.

  • Complexes 1 and 2 are solid compounds. Please, give their melting points.

Complexes 1 and 2 decompose before melting, consequently it is no possible to measure the melting point.

After these corrections the paper may be accepted.

Reviewer 3 Report

The manuscript from the group of Cerrada reports the preparation of a series of gold-carbene complexes containing flavone-functionalized alkynyl ligands. The alkynes and the gold complexes were prepared by established routes and the products were characterized by various spectroscopic techniques. Although the spectroscopic data is sound and confirms identity and purity of the compounds, no structural data is available. A number of studies related to antibacterial activity and cytotoxicity were carried out. Some of the complexes displayed quite high activity and correlations between activity and lipophilicity were examined. The authors have gone a little overboard here and take the very simple view that activity of a substance is solely related to its lipophilicity. While this is undoubtably important, there are of course many other mechanisms in play. This section of the discussion should be turned down a little. The antibacterial studies were carried out using the disk diffusion method at a single concentration. It would have been nice to get some concentration dependent data for better comparison with other metal-based antibiotics. Silver of course is especially relevant here. Have the authors examined the silver-counterparts of the compounds? They would be expected to have even higher antibacterial activity. Overall, the manuscript is of interest and should be published in this journal after minor corrections. In the experimental section there is no reference to the preparation of the ligands. The authors should add a sentence and citation (ref. 46) here.

Author Response

  • The manuscript from the group of Cerrada reports the preparation of a series of gold-carbene complexes containing flavone-functionalized alkynyl ligands. The alkynes and the gold complexes were prepared by established routes and the products were characterized by various spectroscopic techniques. Although the spectroscopic data is sound and confirms identity and purity of the compounds, no structural data is available. A number of studies related to antibacterial activity and cytotoxicity were carried out. Some of the complexes displayed quite high activity and correlations between activity and lipophilicity were examined. The authors have gone a little overboard here and take the very simple view that activity of a substance is solely related to its lipophilicity. While this is undoubtably important, there are of course many other mechanisms in play. This section of the discussion should be turned down a little.

The discussion has been turned down as suggested by the referee

  • The antibacterial studies were carried out using the disk diffusion method at a single concentration. It would have been nice to get some concentration dependent data for better comparison with other metal-based antibiotics. Silver of course is especially relevant here. Have the authors examined the silver-counterparts of the compounds? They would be expected to have even higher antibacterial activity. Overall, the manuscript is of interest and should be published in this journal after minor corrections.

With our purpose of comparing the products among them, rather than with other antibiotics, we used the lowest concentration that showed antibacterial activity in those complexes that effectively inhibited bacterial growth in a wider bacterial spectrum.

We did not synthesize the corresponding silver derivatives, consequently not measurement of their activity was included in the paper. Probably they will exert good values, but it was not the purpose of our research.

  • In the experimental section there is no reference to the preparation of the ligands. The authors should add a sentence and citation (ref. 46) here.

We have included the corresponding reference to 3-hydroxyflavones and the corresponding flavone-derived ligands (La-d) in the experimental section

Round 2

Reviewer 1 Report

In my opinion, after the suggested corrections, the manuscript is suitable for publication in the Pharmaceutics journal.